# Nursing activities and associated workload of nurses in virtual care centres: A multicentre observational study

Jobbe P. L. Leenen[1,2,3☯]*, Jedidja Lok-Visser[1,4☯], Cindy Vollenbroek[4], Henk Sonneveld[5], Thijs Van Houwelingen[5], Gréanne Leeftink[1,4]

**1** Connected Care Center, Isala, Zwolle, The Netherlands, **2** Research Group IT Innovations in Healthcare, Windesheim University of Applied Sciences, Zwolle, The Netherlands, **3** Isala Academy, Isala, Zwolle, The Netherlands, **4** Center for Healthcare Operations Improvement & Research, University of Twente, Enschede, The Netherlands, **5** Research Group Technology for Healthcare Innovations, Research Centre for Healthy and Sustainable Living, University of Applied Sciences Utrecht, Utrecht, The Netherlands

☯ These authors contributed equally to this work.
* j.p.l.leenen@isala.nl

## Abstract

Virtual care centres (VCCs) are novel wards of hospitals and facilitate the provision of remote monitoring and home-based patient care by virtual care nurses. Whereas since the COVID-19 pandemic VCCs have rapidly emerged, there is a lack of insight in virtual care nurses' work and the associated work load. Therefore, the aim of this study was to identify the nursing activities performed in Virtual Care Centers (VCCs) and assess nurses' perceived workload associated with these activities. A multicentre descriptive, observational cross-sectional study was performed. Data collection (February – June 2024) involved three steps: establishing a list of nursing activities, defining and quantifying workload using the NASA-Task Load Index and Analytical Hierarchy Process (AHP), and measuring nursing activity-associated workload by a survey involving 19 virtual care nurses across six VCCs in the Netherlands who had been employed in VCCs for at least one year. Eventually, we identified 21 nursing activities categorized into five areas: education and training (n = 2), development and promotion of new care pathways (n = 4), patient contact (n = 4), clinical decision-making (n = 8), and administration (n = 2). The overall workload was predominantly rated as low to medium, with the development of protocols for new digital care pathways being the most demanding activity. Routine nursing activities, such as patient contact and clinical decision-making, resulted in low to very low workload ratings. In conclusion, we found VCC nurses engage in a broad spectrum of conventional and novel nursing tasks, of which we measured their associated workload using a novel approach integrating NASA-TLX and AHP. The highest associated workload suggest the need for task differentiation and/or additional training to support nurses in managing these high-demand tasks. The VCC model may offer a viable

**Data availability statement:** Data is available at figshare: https://doi.org/10.6084/m9.figshare.28675802.v1.

**Funding:** The author(s) received no specific funding for this work.

**Competing interests:** Jobbe PL Leenen has received an honorarium as a member of the General Care Monitoring Advisory Board of Philips Healthcare.

alternative for nurses experiencing high workloads in conventional wards, potentially alleviating some pressures on nursing staff in traditional healthcare settings, mostly in the shift from physical to mental demand.

### Author summary

In our study, we explored the roles and workload of nurses working in Virtual Care Centers (VCCs), which have become more common since the COVID-19 pandemic. VCCs allow nurses to monitor and care for patients remotely. We aimed to understand what activities these nurses perform and how demanding these tasks are. We conducted a study across multiple centers in the Netherlands, involving 19 nurses who have been working in VCCs for at least a year. We identified 21 different nursing activities, which we grouped into five categories: education and training, developing new care pathways, patient contact, clinical decision-making, and administration. We found that most tasks had a low to medium workload, but creating new digital care protocols was the most challenging. Our findings suggest that while VCCs offer a broad range of nursing activities, the most demanding tasks may require additional training or task differentiation to help nurses manage their workload better. This model could be a good alternative for nurses in traditional settings, potentially reducing their physical workload and shifting more towards mental tasks.

### Introduction

Healthcare systems worldwide are encountering significant challenges. The number of older patients with complex conditions is rising, leading to a substantial increase in healthcare expenditures [1,2]. Additionally, there is a growing shortage of adequately trained healthcare professionals to provide patient-centred care and support [3]. This situation necessitates the development of innovative care models that ensure sustainable, safe, effective, and accessible healthcare delivery [4,5].

One promising model is care delivery in a Virtual Care Centre (VCC). Although there is no standardized definition of a VCC, in this study we define VCCs as a hospital department that specialise in providing remote care to enable remote disease monitoring and facilitate interaction between patients and healthcare professionals without the need for an in-person visit or admission to a healthcare facility [6,7]. This remote care is virtual and delivered through various communication tools such as telephone and video calls, secure messaging platforms, and mobile applications for remote monitoring and connecting patients with healthcare professionals [7,8].

In the Netherlands, the COVID-19 pandemic accelerated the implementation of VCCs in numerous hospitals. These centres have the potential to improve healthcare access, reduce costs, and enhance patient empowerment, making healthcare services more efficient and accessible to a broader population beyond COVID-19

patients [9]. VCCs in the Netherlands deliver a range of hospital-at-home nursing care using information and communication technology, for example focusing on postoperative care, chronic disease management, preventive care, and pre- and rehabilitation care for various specialisms, catering to both inpatients and outpatients.

Virtual care nurses working in VCCs play a pivotal role. These nurses are specialized in providing remote care, which includes offering virtual feedback, guidance, and contact to empower patients to discuss their health conditions and treatment options remotely (e.g., receiving medical advice and instructions, obtaining prescriptions). In our previous qualitative study [10], we found that working in VCCs demands different competencies and roles from nurses due to the extended use of remote care requiring more technical skills, clinical decisions making, lack of physical patient contact, the high innovative and evolving nature of remote care. We therefore hypothesize that that these changes may have both positive and negative impacts on nurses' job satisfaction, workload and retention. However, it is unknown whether this is related and can be generalized to other VCCs.

Virtual care nurses typically work in less physically demanding environments compared to traditional bedside nurses. This potentially reduces physical issues such as back pain and fatigue, which are common among hospital-based nurses and which may potentially lead to (prolonged) sick leave and/or job changes [11]. However, virtual care nurses may face different types of stress, such as technological challenges, isolation from colleagues, and the need for effective communication skills to manage patient care remotely. These factors contribute to mental and emotional stress, even though they may not experience the same physical demands as traditional nurses [12,13].

Given the potential changes in nursing work, coupled with the rapid implementation VCCs, the organization of VCCs varies significantly across hospitals and practices, often lacking a standardized, evidence-based approach [14]. There is a lack of clarity regarding the specific nursing activities and associated workload within these centres. Understanding the range of nursing activities and associated workload is crucial for improving nurses' working conditions, as well as for developing training and education, and for retaining nurses in this upcoming nursing field. Therefore, this study aims to 1) identify the nursing activities performed in VCCs and 2) assess nurses' perceived workload associated with these activities.

## Materials and methods

### Study design

A multi-centre descriptive, observational cross-sectional study was performed in February-June 2024. This study is reported in concordance with the STrengthening the Reporting of OBservational studies in Epidemiology (STROBE) [15].

### Setting

The study was conducted in VCCs within six hospitals in the Netherlands. These included five top clinical teaching hospitals and one university medical centre. The primary focus of these VCCs was to utilize digital technologies for delivering medical services virtually and remotely, as well as providing intravenous infusion therapy at home (e.g., Outpatient Parenteral Antimicrobial Therapy and ketamine infusion therapy) without restriction to specific diseases or specialist care pathways. All VCCs were established during the COVID-19 pandemic, between 2020 and 2022.

The VCCs varied in several aspects, including their size, the number of patients receiving remote care, the number and types of care pathways included, and the number of registered nurses working in each VCC. Given the evolving nature of VCCs and the variability in their characteristics, it was not feasible to collect detailed data per hospital.

### Participants

Participants approached for this study were virtual care nurses employed in the VCCs of six hospitals in the Netherlands. The inclusion criteria were: current employment within the VCC of their respective hospital for a minimum of one year and

the ability to read and understand the Dutch language. Participants were approached via email after obtaining approval from the manager of each participating hospital. All eligible participants from all six hospitals were invited to complete the survey.

### Data collection

We define work as the process of transforming inputs into outputs, with activities being the units that contribute to producing the final output [16]. This study focuses on the content dimension to define activities, making it broadly applicable and suitable for examining the activities performed by VCC nurses. To facilitate the examination of workload using an online questionnaire, we undertook three steps: 1) establishment of a list of nursing activities working in a VCC, 2) definition and quantification of workload and 3) nursing activity-associated workload was operationalized.

**Step 1: Nursing activities.**  For the development of the activities list, three VCC nurses from Hospital A, who have extensive experience in providing remote care, were involved. These nurses were selected to develop a general and concise activities list that accurately represents the profession of a VCC nurse.

The activities list was developed through a three-round process. Initially, the research team (JL, JLV, CV) created a preliminary concept based on themes from a previous qualitative study on the work of VCC nurses and a study on nurses working with remote care in home care settings [10]. To account for activities that are approached either ad-hoc or in a planned manner, this differentiation was made in the activity descriptions, as previous studies indicated that unplanned activities might result in a higher perceived workload [17,18] (S1 Table). In the second round, the activities list was discussed and verified by the three VCC nurses and JL, JLV and CV. The list was further refined using the seven Canadian Medical Education Directions for Specialists (CanMEDS) roles from the Dutch educational profile for bachelor nurses (Bachelor Nursing 2030). Finally, the list was discussed with the nurses again but no additions were made, and consensus was reached together with the research team based on verbal agreement. In the survey, nurses of the other hospitals were able to indicate whether they performed each activity and could rate the activity as 'not applicable' if they did not perform the task.

**Step 2: Workload definition and quantification.**  Workload is a concept that has been described in multiple ways. We chose a nurse-based subjective workload which is defined based on Hart et al. [19] definitions as the subjective effect on a nurse from performing activities. To measure subjective workload, we used the NASA-Task Load Index (TLX) [19]. The NASA-TLX is a validated, employee-based questionnaire to measure subjective workload and is commonly used in studies in nursing [20–23]. The rationale of the NASA-TLX is that workload consists of six workload dimensions, that can be measured from two perspectives, the extent of which the workload dimension is present in a task, and the subjective weight of that dimension. E.g., if a task is physical very demanding, but a nurse does not perceive the physical demand dimension as an important part of workload, the subjective workload of this task for this dimension is still relatively low for this nurse.

The six workload dimensions are: mental demand, physical demand, temporal demand, effort, frustration level, and performance. Together, these variables represent the total workload of activities. The NASA-TLX assigns a quantitative workload measure to activities based on the weights of these six dimensions and the extent to which each dimension is present in each activity.

Weights were assigned to the workload dimensions through pairwise comparisons by means of the Analytical Hierarchy Process (AHP). Combining the NASA-TLX with the AHP provides more accurate weights in comparison with the conventional method within the NASA-TLX, as proposed by Virtanen et al. [24]. The AHP provides the option to assess variables as equally important and therefore include all six variables in the workload, which is not possible in the conventional method [24]. The aim of these pairwise comparisons is to assign a numerical weighted importance to all six workload dimensions, per nurse, resulting in a sum of 1. To come to this, according to the AHP, participants compared two variables at a time to determine which was more important on a cardinal scale ranging from -9–9 [25]. A total of 15 comparisons

were made to establish the relative importance of the dimensions, which were then used to determine their weights. An example of the outcome could be mental demand: 0.4, physical demand: 0.05, temporal demand: 0.1, effort: 0.2, frustration: 0.2, performance: 0.05.

**Step 3: Nursing activity-associated workload.** To measure the nurses' perceived workload for each of the identified activities, virtual care nurses employed in the VCCs of the six hospitals were surveyed (S1 Appendix). In this survey, the nurses first made pairwise comparisons, as described in step 2. Then, participants indicated the extent to which each variable contributed to each workload dimension for each activity using a scale from 0 to 100, with increments of 5 points. A score of 0 indicates that the workload dimension is not present in this activity, while a score of 100 indicates maximum workload for that dimension. This method allowed the six workload dimensions to collectively represent the total workload experienced for a specific activity. According to the NASA-TLX, the total workload of an activity was calculated by summing the scores multiplied by their respective weights (Fig 1). See S2 Appendix for a calculation example.

**Other parameters.** The demographic data was gathered to get insight into the composition of the sample: 1) age, 2) gender, 3) job function, 4) education level, 5) affiliated hospital, 6) years active as a nurse, 7) years active as a virtual care provider and 8) hours working per week.

## Data analysis

The nursing activities list was generated based on the three rounds reaching verbal agreement from all nurses and the research team. For inclusion in the analysis of the survey, participants had to complete the pairwise comparisons of the workload dimensions and at least 1/3 of the activity associated workload. All data were analysed using IBM SPSS Statistics (*version 29.0.2.0; IBM, Armonk, NY, USA*). A *p*-value of <0.05 was considered as statistical significant.

First, the demographic data and the performance of the activities list of VCC nurses are evaluated by descriptive statistics using frequency and percentage for categorical and ordinal data and means and standard deviations for normally distributed continuous data. Second, total workload was determined per participant, per activity, by summing the scores multiplied by their respective weights (Fig 1) [24]. Weights of the workload dimensions were presented as median and interquartile ranges and nursing activity-associated workload (range 0–100) was tested on statically significance differences between activities using the Wilcoxon-Mann-Whitney test. To control for multiple comparisons (n = 210 comparisons), the Bonferroni correction was applied, adjusting the significance level to p < 0.001. The mean workload per activity was categorized into five levels: very low (0–9), low (10–29), medium (30–49), high (50–79), and very high (80–100) [26].

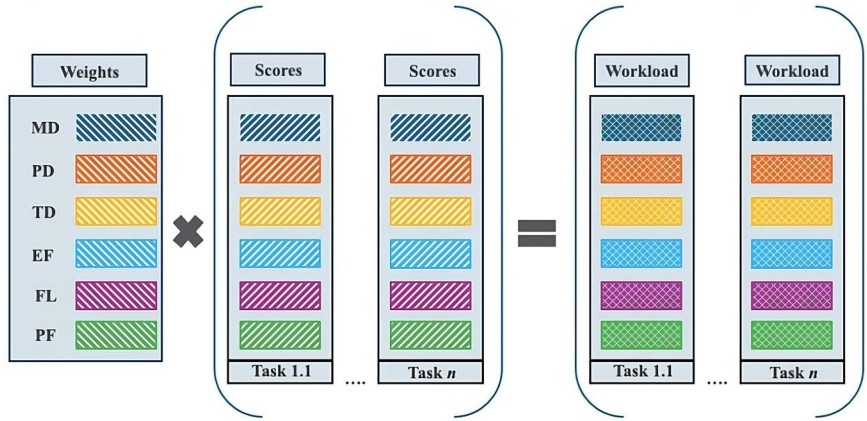

**Fig 1. Calculation of nursing activity-associated workload, per activity.** MD: Mental demand, PD: Physical demand, TD: Temporal demand, EF: Effort, FL: Frustration.

For a validity check of the pair-wise comparisons of workload dimensions by the nurses, a consistency ratio (CR) was derived to what extent respondents were consistent in assigning weights, according to the AHP methodology. A CR of >0.1 implies slight inconsistencies may be present (see S3 Appendix). [27]

### Ethical considerations

Both the Medical Ethics Committee of the Isala Hospital reviewed the protocol (protocol nr. 20240419) and BMS ethical committee/ Domain Humanities & Social Sciences (protocol nr. 240653) and declared that the Medical Research Involving Human Subjects Act did not apply for this study. This study was conducted in accordance with the Declaration of Helsinki. Informed consent was obtained from all participants in the study.

## Results

### Nurse characteristics

Nineteen nurses were included of which 6% (n = 2) completed less than 1/3 of the activity associated workload and were excluded from analysis for activities-oriented workload. 95% (n = 18) of the nurses were female with a mean age of 39 years old (SD:11) and work experience in the VCC of 2.3 (SD: 0.9) years. All demographic data are summarised in Table 1.

### Activities of VCC nurses

After the first round, 16 activities divided over five categories were defined in the first round (S1 Table). After the second round, consensus was reached and a total of 18 activities were defined and divided over five categories: 1) Education and training, 2) Development and promotion of new care pathways, 3) Patient contact, 4) Clinical decision-making, and 5) Administration (Table 2). Three activities were categorized based on whether they were approached in an ad-hoc or planned manner.

**Table 1. Overview of demographical data of VCC nurses (n = 19).**

| Characteristics | |
|---|---|
| **Age, mean (SD)** | 39 (11) |
| **Sex, female, n (%)** | 18 (95) |
| **Education level, n (%)** | 3 (16) |
| In-service education (NLFQ-4) | 10 (53) |
| Bachelor degree | 4 (21) |
| Bachelor degree with specialisation | 1 (5) |
| Master degree | 1 (5) |
| Other | |
| **Hours working per week, mean (SD)** | 27.7 (7.6) |
| **Years active as care provider, mean (SD)** | 19.7 (12.6) |
| **Years active as virtual care nurse, mean (SD)** | 2.3 (0.9) |
| **Hospital, n (%)** | 8 (42) |
| Hospital A (Top Clinical Teaching Hospital) | 1 (5) |
| Hospital B (Top Clinical Teaching Hospital) | 5 (26) |
| Hospital C (Top Clinical Teaching Hospital) | 2 (11) |
| Hospital D (Top Clinical Teaching Hospital) | 1 (5) |
| Hospital E (Top Clinical Teaching Hospital) | 2 (11) |
| Hospital F (University Medical Center) | |

**Table 2. Overview of nursing activities in virtual care centres, performance rate and activity-associated workload (n = 19).**

| Activities category | Activity description | Activity performed, n (%) | Workload rating, median (IQR) | Workload category |
|---|---|---|---|---|
| 1. Education and training | 1. Organise and develop education and training for new care pathways or for introduction of new colleagues | 18 (95) | 30.1 (10.9-54.1) | Medium |
| | 2. Attending education and training to maintain and expand knowledge | 18 (95) | 22.0 (7.6-47.4) | Low |
| 2. Development and promotion of new care pathways | 1. Develop protocols (work process and used technology) of new digital care pathways | 17 (90) | 49.8 (36.4-56.2) | Medium |
| | 2. Testing, evaluation, optimisation, and quality assurance of protocols (work process and technology used) | 18 (95) | 34.9 (12.9-50.1) | Medium |
| | 3. Active participation in quality assurance systems to improve the quality of care (e.g., VIM committee, quality monitoring and/or improvement of protocols and/or digital platforms) | 16 (84) | 21.2 (7.8-44.5) | Low |
| | 4. Ambassadorship of virtual care (Communication towards external stakeholders to promote digital care). | 12 (63) | 18.5 (12.7-49.2) | Medium |
| 3. Contact with patients | 1.1. Planned remote patient counselling and coaching (e.g., for self-management, increased adherence, and/or psychological support.) | 17 (90) | 21.1 (5.7-39.0) | Low |
| | 1.2. Ad-hoc remote patient counselling and coaching (e.g., for self-management, increased adherence, and/or psychological support.) | 18 (95) | 11.7 (6.9-43.5) | Low |
| | 2.1. Planned remote communication with the patient to enable clinical decision-making (e.g., to verify data (e.g., measurement value) with the patient and/or give instructions on technology/instrument use). | 18 (95) | 8.8 (5.7-40.3) | Very low |
| | 2.2. Ad-hoc remote communication with the patient to enable clinical decision-making (e.g., for verification of data (e.g., measurement value) with the patient and/or give instructions on technology/instrument use). | 18 (95) | 16.1 (5.7-43.2) | Low |
| | 3. Performing nursing procedures (e.g., IV puncturing). | 8 (42) | 17.9 (7.6-49.6) | Low |
| 4. Clinical decision making | 1. Assess information from telemonitoring only via data platforms (e.g., Luscii, Curavista, SanaNet, HiX). | 19 (100) | 12.7 (4.8-42.8) | Low |
| | 2. Assess information only via patient contact (e.g., phone call). | 16 (84) | 8.1 (3.8-25.8) | Very low |
| | 3. Assessing information from data platforms as well as patient contact. | 18 (95) | 11.0 (4.7-23.7) | Low |
| | 4. Decision-making based on protocols. | 19 (100) | 6.8 (5.1-33.0) | Very low |
| | 5. Decision-making based on consultation with a hospital colleague (e.g., doctor, nursing specialist, and/or protocol owner). | 18 (95) | 17.9 (5.2-36.2) | Low |
| | 6. Coordinate and realise specialised hospital-at-home care (e.g., OPAT) | 10 (53) | 12.3 (5.7-37.1) | Low |
| | 7.1. Planned handling of notifications from the monitoring platform used. | 18 (95) | 8.7 (3.7-41.9) | Very low |
| | 7.2. Ad-hoc handling of notifications from the monitoring platform used. | 17 (90) | 10.4 (4.9-36.8) | Low |
| 5. Administration | 1. Record keeping of care performed (e.g., in Luscii, Curavista, SanaNet, HiX et al.) | 19 (100) | 8.9 (6.1-25.1) | Very low |
| | 2. Perform Human Resource Management (HRM) duties (e.g., scheduling, HR administration) | 8 (42) | 10.2 (5.8-28.0) | Low |

n: frequency. IQR: interquartile range. Workload rating range: 0–100. Work load category: very low (0–9), low (10–29), medium (30–49), high (50–79), and very high (80–100).

Following consensus, the activities were scored by all VCC nurses. From the activity list, Activity 4.1 *assessing information from telemonitoring only via a data platform*; Activity 4.4 *decision-making based on protocols*; and Activity 5.1 *decision-making based on consultation with a hospital colleague* were performed by all respondents. In contrast, Activity 3.3 *performing nursing procedures* and Activity 5.2 *performing Human Resource Management (HRM) duties* were performed the least, with only 42% (n = 8) of respondents engaging in these activities.

## Workload

The median weights (with interquartile ranges) for the workload dimensions rated by the nurses were as follows: Mental Demand (MD) at 0.227 (IQR 0.153-0.314), Frustration Level (FL) at 0.193 (IQR 0.167-0.251), Performance (PF) at 0.158 (IQR 0.099-0.203), Temporal Demand (TD) at 0.161 (IQR 0.107-0.221), Effort (EF) at 0.125 (IQR 0.100-0.164), and Physical Demand (PD) at 0.067 (IQR 0.042-0.142). For n = 7 (37%) nurses their CR-score was below the limit of 0.1, implying consistency.

## Nursing activity associated workload

For the categories, median workload for *education and training* was 28.1 (IQR: 11.2–48.4), for *development and promotion of new care pathways* was 32.9 (IQR: 24.1–49.0), for *contact with patients* was 15.6 (IQR: 6.4–38.7), for *clinical decision making* was 14.8 (IQR: 6.0–21.4) and for *administration* was 8.9 (IQR: 7.3–24.1). Per activity, the nursing activity-associated workload was rated as Very Low (n = 6), Low (n = 10) and Medium (n = 5), with no activities as High or Very high (Table 2; Fig 2). Here, Activity 2.1, i.e., *developing protocols for new digital care pathways, including work processes and used technology*, had the highest median workload score of 49.8 (IQR = 36.4-56.2). Conversely, Activity 4.4, i.e., *decision-making based on protocols*, had the lowest median workload score of 6.8 (IQR: 5.1-33.0). When comparing the individual workloads, Activity 2.1, i.e., *developing protocols for new digital care pathways, including work processes and used technology*, was significantly higher than six other activities: Activity 1.2 *Attending education*, Activity 3.1.1 *Planned patient counselling and coaching*, Activity 3.2.1 *Planned patient communication for clinical decision making*, Activity 4.1 *Assessing information from data platforms*, Activity 4.3 *Assessing information from data platforms as well as patient contact*, and Activity 4.7.2 *ad-hoc handling of notifications* (S2 Table). There were no significant differences between ad-hoc and planned activities (S3 Table).

## Discussion

Our study aimed to identify the nursing activities performed in VCCs and assess nurses' perceived workload associated with these activities using the NASA-TLX combined with AHP. We identified a list of 21 nursing activities, categorized into five main areas: education and training, development and promotion of new care pathways, contact with patients, clinical decision making, and administration. Notably, the majority of activities such as assessing information from telemonitoring, decision-making based on protocols, and consultation with hospital colleagues were generally performed by all nurses, whereas tasks like performing physical nursing procedures and HRM duties were less common. The workload analysis indicated that the overall nursing activity-associated workload was predominantly rated as low (n = 18) to medium (n = 3), with no activities classified as high or very high. Specifically, the development of protocols for new digital care pathways was identified as the activity with the highest workload, while decision-making based on protocols was the least demanding

### Comparison with prior research

The list of nursing tasks predominantly consisted of tasks related to regular nursing work. These findings are in line with prior studies where care coordination, documentation, effective communication, patient assessment, - diagnosis and clinical decision-making, patient safety and leadership also were found as important competencies [28,29]. However, education and training, as well as the development and promotion of new care pathways, appear to play a substantial role in the work of VCC nurses (28% of all tasks). This emphasis on education and development can be attributed to the innovative and evolving nature of their roles, where continuous education is essential for delivering high-quality care [10]. Additionally, we observed that VCC nurses play a crucial role in organizing this education, which is often centrally managed in hospitals by staff with an educational background. This raises important questions about whether these responsibilities

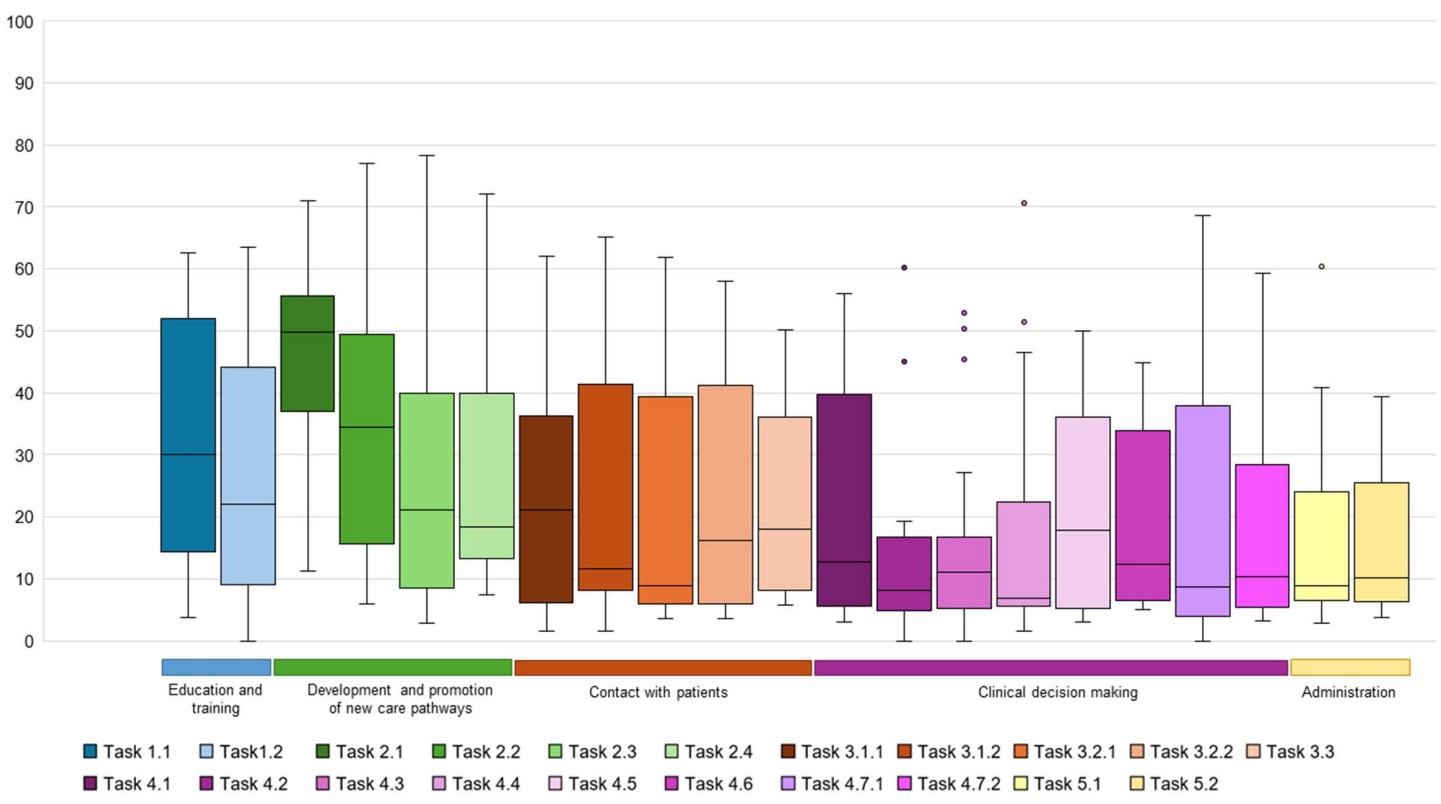

**Fig 2. Distribution in median workload per activity in Virtual Care Centres.** Blue: Education and training; Green: Development and promotion of new care pathways, Orange: Contact with patients; Purple: Clinical decision making; Yellow: Administration.

align with the traditional scope of nursing work. Should nurses acquire additional didactic skills to better organize education, or should they be supported by educational specialists within or outside the hospital [30].

Furthermore, our study revealed that VCC nurses overall rated their workload as very low or low. Although the mental demand was also the highest workload factor for VCC nurses in comparison to prior studies, it seems this workload is generally lower than conventional and Intensive Care Unit nurses working in clinical care [20]. Specifically, it appears to be lower in physical demand [21,22]. This contrast is likely attributable to the nature of VCC work, which predominantly involves remote patient interactions and remote monitoring, requiring less physical exertion than traditional hands-on patient care [21,23]. Consequently, the reduced physical demands may explain the lower workload ratings by VCC nurses, highlighting a significant shift in the physical aspects of nursing tasks between these two environments. Conversely, the reduced physical activity associated with sedentary work may contribute to the development of back pain issues. Also, we found VCC nurses have a pivotal role in the development and promotion of new care pathways. The development of protocols, which often include a substantial technological component, their practical testing, and continuous improvement seem to be integral parts of VCC nurses' responsibilities. These tasks are particularly demanding due to their technical complexity, the need for precise coordination among multidisciplinary teams, and the continuous adaptation to new technologies. While these responsibilities are crucial for ensuring high-quality care, it remains uncertain whether nurses can perform these tasks effectively and whether they perceive them as appropriate within their scope of duties. This task causes the highest workload (medium) among all tasks, probably due to the aforementioned complexities. Therefore it seem to be important to support or educate nurses in this task to decrease workload, as high workload

is associated with higher nursing burnout and retention [23]. Additionally, we noted that the nurses act as ambassadors in the implementation of virtual care, which aligns with current trends in nursing leadership and autonomy in implementation of novel (virtual) care pathways [29,31]. However, this is a relative novel role for nurses which also suggest additional education is needed for instance in project management or nurse leadership, or may advocate for task allocation within the nurses or to other disciplines.

The two tasks of developing protocols (work process and used technology) of new digital care pathways and testing, evaluating, optimising, and assuring quality of protocols (work process and technology used) tasks seem to generate more workload (medium) in comparison to the other tasks where we found that they generate very low or low workload experienced by nurses. This could be because the other task categories (e.g., clinical decision making and patient contact) align with conventional nursing work and/or what they learned during their training. Especially because VCC nurses generally had relevant prior work experience in nursing before working in the VCC. This could have helped to gain experience and become familiar with these tasks categories. Despite this, the use of technology by the VCC nurses does not seem to cause a substantial high workload, which might be different for conventional nurses [32,33]. This could be because VCC nurses may naturally be more technology-minded than the majority of the nursing workforce, which is often slower to adopt new technologies and innovations [34]. The integration of technology in VCCs requires nurses to engage in complex tasks such as developing and optimizing digital care protocols, which involve significant technical complexity and coordination among multidisciplinary teams. These tasks demand a high level of technological proficiency and adaptability, which VCC nurses seem to possess to a certain extent. Moreover, the continuous adaptation to new technologies and the need for precise coordination highlight the importance of ongoing education and support for VCC nurses.Considering our method, combining the NASA-TLX with AHP is a novel approach, to the authors' knowledge documented once in literature [20]. This integration leverages the strengths of both tools to provide a comprehensive analysis of workload dimensions. The NASA-TLX is well-regarded for its ability to measure perceived workload across multiple dimensions, while AHP offers a structured technique for organizing and analysing complex decisions [19,24,25]. This combination allows for a more nuanced understanding of workload, particularly in complex environments like VCCs. In the conventional NASA-TLX, the weights are assigned based on ranking, meaning that dimensions cannot be valued equally, and that the lowest-ranked dimension is not included in the analysis. Although this combination provides a detailed analysis, it may also exaggerate extreme scores. For instance, VCC nurses report low physical demands in their work, leading to lower scores in that dimension. This should be carefully considered when interpreting the results, as it may skew the perceived workload in certain areas. Additionally, inconsistencies in participants' responses indicate potential issues with the AHP's subjective pairwise comparisons. Despite these challenges, the combined use of NASA-TLX and AHP enhances the ability to identify specific workload components and their relative importance, which is beneficial for targeted interventions and improvement of nurses' work.

## Limitations

Several limitations should be addressed in the interpretation of our results. First, while the inclusion of six centres adds breadth to the study and there is no sight on the total population of VCC nurses in the Netherlands (which may be small given the novelty of VCCs), the overall sample remains limited (19 participants). The study was conducted in six VCCs with varying sizes, patient numbers, care pathways, and technological setups. This variability could affect the generalizability of the findings, as differences in technological infrastructure and patient populations may lead to distinct experiences and workloads for nurses across different centres. However, due to the dynamic and heterogeneous nature of VCCs, obtaining consistent and comprehensive data across all hospitals was not feasible. Future research should aim to collect and analyse such detailed data to facilitate in-depth comparison of VCC characteristics. Additionally, workload may be influenced by factors such as age, sex, or other contextual variables, which were not fully captured in this study due to the sample size constraints [19,23].

Second, workload can evolve due to factors such as the total amount of activities per shift or simultaneously performing two activities, which can be more demanding than the combined effort of completing each task individually. Additionally, completing this survey at a specific moment in time can influence results [35,36]. This variability may also be seasonal, with higher demands often observed during periods like the flu season. Therefore, the timing of these assessments can significantly impact how tasks and workload are perceived and reported. Although the survey was anonymized, socially desirable responses may have influenced the study results on perceived workload as nurses might have reported lower workload levels to align with perceived expectations or norms.

Third, the activities list used in this study was developed through input from a small group of experienced VCC nurses of one hospital and refined via consensus. Although this hospital is one of the frontrunners in the field of VCCs, it may not encompass the full range of activities or account for variations in practice across different VCCs. However no missed activities in open text field mentioned by nurses from other five hospitals. Additionally, our approach to identifying nursing activities involved a three-step pragmatic method, including consultations and interviews with nurses and the research team. These consultations were analysed through a systematic process of discussion for each task and oral agreement to reach consensus. While this method provided valuable insights, it lacks the rigor of more formal data collection and analysis methods, such as a Delphi round.

Lastly, although the NASA-TLX and AHP provide a more valid method, they may also exaggerate extreme scores. Future studies should consider to apply this combined methodology in different healthcare settings to validate this method further and compare and contrast workload perceptions across various contexts. This would provide a broader perspective on the applicability and robustness of the NASA-TLX and AHP combination. Additionally, our study showed that only 7 (37%) of nurses were consistent in their pair-wise comparisons of the workload dimensions. This inconsistency should be taken into account when interpreting the results, as it suggests that consistently comparing workload dimensions is a complex task for nurses. The inconsistency in pairwise comparisons may be attributed to several factors, including the subjective nature of workload assessment, the cognitive load involved in making multiple comparisons, and potential variations in individual perceptions of workload dimensions. Future research should explore strategies to improve the consistency of pairwise comparisons, such as simplifying the comparison process or providing real-time feedback when filling in the survey.

## Implications

Although most nursing activities were associated with a low to medium workload, addressing the higher workload levels of VCC nurses' activities is crucial for several reasons. One potential strategy is to implement task differentiation within VCCs. This approach can help gaining expertise in certain activities (i.e., developing new care protocols) and distribute the workload more evenly among staff, preventing burnout and enhancing overall efficiency [37]. Additionally, it is important to reassess whether certain tasks, such as protocol development and technological implementation, should fall within the scope of nursing roles or require specialized staff. This reassessment can ensure that nurses are not overburdened with tasks that may be outside their traditional scope of practice, thereby improving job satisfaction and retention. By clearly defining roles and responsibilities, VCCs can optimize task allocation and enhance the overall effectiveness of the care provided.

The overall lower experienced workload in VCCs, compared to traditional nursing departments, suggests that VCCs could play a significant role in improving nurse retention. Nurses experiencing high workload may find VCC roles more sustainable and less exhausting. Consequently, VCCs could offer a viable solution for retaining experienced nurses who might otherwise leave the profession due to high (physical) demands [38]. This retention of skilled nurses is crucial for maintaining a competent workforce and ensuring sustained, high-quality patient care [39].

Integrating VCC activities into nursing curricula and offering education and training on digital health from the outset ensures that these tasks become part of the professional identity of nurses, thereby better preparing them to manage

the unique demands of providing remote care in VCCs. This preparation may result in a better fit for this work, potentially reducing workload and increasing job satisfaction. Given the specific context and demands of VCCs, the activities identified in this study can serve as a starting point for developing training and educational programs. As the number of VCCs continues to grow and the scope of care expands, it is crucial to prepare future nurses for these roles during their education. This proactive approach will ensure that nursing graduates are equipped with the necessary skills and knowledge to thrive in virtual care environments [40].

## Conclusion

This study provides valuable insights into the nursing activities performed in VCCs in the Netherlands and their associated workload using a novel approach integrating NASA-TLX and AHP. The results show that VCC nurses engage not only in a broad spectrum of conventional nursing tasks, such as patient contact and clinical decision-making, but also in new responsibilities, including the development of educational programs, the creation of digital care protocols in collaboration with other healthcare professionals, and acting as ambassadors for remote hospital care. Notably, the highest workload was associated with these new tasks, highlighting the complexity and demanding nature of these responsibilities in an innovative setting like the VCC. This suggests that task differentiation or additional training may be necessary to support VCC nurses in managing these high-demand tasks. In contrast, majority of nursing activities resulted in low to very low workload ratings, indicating that the virtual ward model may offer a viable alternative for nurses experiencing high workloads in conventional wards and emphasizes the potential of VCCs to alleviate some of the pressures on nursing staff in traditional healthcare settings, while also underscoring the need for tailored support in the unique and evolving roles within remote care.

## Supporting information

**S1 Appendix.  survey used in the study**
(DOCX)

**S2 Appendix.  calculation example for nursing activity-associated workload**
(DOCX)

**S3 Appendix.  calculation of consistency ratio**
(DOCX)

**S1 Table.  The nursing activities list per round.**
(DOCX)

**S2 Table.  Wilcoxon-Mann-Whitney test on tasks.**
(DOCX)

**S3 Table.  Wilcoxon-Mann-Whitney test on workload ad-hoc and planned activities.**
(DOCX)

## Acknowledgments

The authors would like to thank all participating nurses, Aenne Merkx (manager of VCC Isala) and all other managers of the VCCs.

## Author contributions

**Conceptualization:** Jobbe P. L. Leenen, Jedidja Lok-Visser, Cindy Vollenbroek, Thijs van Houwelingen, Gréanne Leeftink.

**Data curation:** Jedidja Lok-Visser, Cindy Vollenbroek.

**Formal analysis:** Jedidja Lok-Visser, Cindy Vollenbroek.

**Investigation:** Jobbe P. L. Leenen, Jedidja Lok-Visser, Cindy Vollenbroek, Gréanne Leeftink.

**Methodology:** Jobbe P. L. Leenen, Jedidja Lok-Visser, Cindy Vollenbroek, Gréanne Leeftink.

**Project administration:** Jedidja Lok-Visser, Cindy Vollenbroek, Gréanne Leeftink.

**Resources:** Jobbe P. L. Leenen.

**Supervision:** Jobbe P. L. Leenen, Jedidja Lok-Visser, Gréanne Leeftink.

**Validation:** Jobbe P. L. Leenen, Jedidja Lok-Visser, Henk Sonneveld, Thijs van Houwelingen.

**Visualization:** Jedidja Lok-Visser.

**Writing – original draft:** Jobbe P. L. Leenen.

**Writing – review & editing:** Jobbe P. L. Leenen, Jedidja Lok-Visser, Cindy Vollenbroek, Henk Sonneveld, Thijs van Houwelingen, Gréanne Leeftink.

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
