## [Decision Letter · Decision Letter 0]

28 Feb 2025

Response to Reviewers
Revised Manuscript with Track Changes
Manuscript
**Journal Requirements:**

1. We have amended your Competing Interest statement to comply with journal style. We kindly ask that you double check the statement and let us know if anything is incorrect.

2. Please provide an Author Summary. This should appear in your manuscript between the Abstract (if applicable) and the Introduction, and should be 150–200 words long. The aim should be to make your findings accessible to a wide audience that includes both scientists and non-scientists. Sample summaries can be found on our website under Submission Guidelines:

https://journals.plos.org/digitalhealth/s/submission-guidelines#loc-parts-of-a-submission

3. In the online submission form, you indicated that “Data is available upon reasonable request at the corresponding author.”.

3. Uploaded as supplementary information.

**Additional Editor Comments (if provided):**
**Reviewers' Comments:**

**Comments to the Author**

1. Does this manuscript meet PLOS Digital Health’s publication criteria?

Reviewer #1: Yes

Reviewer #2: Yes

Reviewer #3: Yes

Reviewer #4: Yes

2. Has the statistical analysis been performed appropriately and rigorously?

Reviewer #1: Yes

Reviewer #2: Yes

Reviewer #3: Yes

Reviewer #4: Yes

3. Have the authors made all data underlying the findings in their manuscript fully available (please refer to the Data Availability Statement at the start of the manuscript PDF file)?

Reviewer #1: Yes

Reviewer #2: Yes

Reviewer #3: No

Reviewer #4: Yes

4. Is the manuscript presented in an intelligible fashion and written in standard English?

Reviewer #1: Yes

Reviewer #2: Yes

Reviewer #3: Yes

Reviewer #4: Yes

Reviewer #1: Main Comments:

1. Clarity of Objectives: The rationale for using NASA-TLX and AHP together should be better explained. Why was this combination chosen, and how does it uniquely contribute to the analysis?

2. Sample Size and Generalizability: The sample size is small, and variability in VCC characteristics may limit the generalizability of findings. This limitation should be explicitly discussed in the manuscript.

3. Workload Insights: Tasks such as protocol development were identified as most demanding but lack detailed discussion. Provide more granular insights into what makes these tasks challenging (e.g., technical complexity, coordination requirements).

4.Ambassadorship and Education Roles: Reassess whether these tasks should fall within the scope of nursing roles or require specialized staff. This is an important point for broader discussions on task allocation.

5. Comparative Analysis: The manuscript claims that VCC work is less physically demanding than traditional nursing roles but lacks comparative data. Include evidence or references to support this claim.

6. Figures and Tables: Consider improving the visual representation of workload categories in Figure 2 for better comparison between activities.

Minor Comments:

1. Address grammatical and formatting inconsistencies.(e.g., "the highest associated workload is suggest the need"). Proofreading is necessary.

2. Strengthen the discussion of methodological limitations, particularly around the inconsistency in pairwise comparisons.

Reviewer #2: The study design is robust, using validated workload assessment tools (NASA-TLX, AHP). However, the workload measurement method could be explained more clearly for those unfamiliar with it. Discussion provides meaningful interpretation but has some redundancy. The workload findings and technology-related insights are interesting but could be more critically analyzed.

Reviewer #3: Thank you for giving me the opportunity to review this manuscript. The paper is well and quite easy to follow. I had a serious problem with the sample size but the authors acknowledged that as a limitation. I have few suggestions to offer except the following

1.The data analysis part is quite short and that will be difficult for other researchers to replicate in other contexts. The authors should consider expanding on the data analysis approach, particularly the process in which they identified the nursing activities. If the nurses were consulted/interviewed, how did the authors analyzed that data to arrive at the nursing activities or how did the data inform the revisions of those nursing activities. These processes need to be made explicit.

2.The authors talk of remote and virtual care and I am not sure if there any differences between the two

Reviewer #4: See comments/suggestions

Abstract

1. Double check for clarity: The highest associated workload is suggest the need...

2. Indicate breakdown of 21 nursing activities for each area.

Introduction

3. To better organize your results later, it is good to explicitly write your research objectives at the end of the introduction.

Methods

4. Possible to create a table summarizing how different each VCCs are? Like a table comparing each VCC based on the aspects your mentioned.

5. What do you mean by this statement: "nurses of the other hospitals were given the opportunity to indicate whether they performed each activity."?

6. Was consensus about activities reached through qualitative (verbal agreement) or quantitative means (computing interrater reliability?)

7. Need citations to justify that TLX is commonly used in nursing studies.

8. Need citations to support combining TLX with AHP. Has this been done before? Similar to the use of CR also. Also it is novel to do that (based on the discussion section), was there any inspiration from previous work to do this approach?

9. Include the survey form used in the study.

Results

10. In table 2, why assign 1.1, 1.2 for some items instead of separate counts (1,2,3,4,5...)?

11. For Figure 2, I feel that this needs to be sorted based on the most to lowest workload.

12. Can you clarify this?: "For n=7 (37%) nurses their CR-score was below the limit of 0.1, implying consistency"

13. Is it possible to compute a combined workload rating at the category level?

Discussion

14. I think it is important to account for social desirability when the participants judge their workload. Perhaps a discussion on that can be made in the limitations section.

**Do you want your identity to be public for this peer review?** For information about this choice, including consent withdrawal, please see our Privacy Policy

Reviewer #1: No

Reviewer #2: No

Reviewer #3: No

Reviewer #4: No

**Figure resubmission:****Reproducibility:** To enhance the reproducibility of your results, we recommend that authors of applicable studies deposit laboratory protocols in protocols.io, where a protocol can be assigned its own identifier (DOI) such that it can be cited independently in the future. Additionally, PLOS ONE offers an option to publish peer-reviewed clinical study protocols. Read more information on sharing protocols at https://plos.org/protocols?utm_medium=editorial-email&utm_source=authorletters&utm_campaign=protocols

---

## [Decision Letter · Decision Letter 1]

18 Jul 2025

Nursing activities and associated workload of nurses in virtual care centres: a multicentre observational study

PDIG-D-25-00005R1

Dear Dr. Leenen,

We're pleased to inform you that your manuscript has been judged scientifically suitable for publication and will be formally accepted for publication once it meets all outstanding technical requirements.

Within one week, you'll receive an e-mail detailing the required amendments. When these have been addressed, you'll receive a formal acceptance letter and your manuscript will be scheduled for publication.

An invoice for payment will follow shortly after the formal acceptance. To ensure an efficient process, please log into Editorial Manager at https://www.editorialmanager.com/pdig/ click the 'Update My Information' link at the top of the page, and double check that your user information is up-to-date. For billing related questions, please contact billing support at https://plos.my.site.com/s/.

Kind regards,

Haleh Ayatollahi

Section Editor

PLOS Digital Health

Additional Editor Comments (optional):

Reviewers' comments:

Reviewer's Responses to Questions

**Comments to the Author**

Reviewer #2: All comments have been addressed

Reviewer #3: All comments have been addressed

Reviewer #4: All comments have been addressed

publication criteria?

Reviewer #2: Yes

Reviewer #3: Yes

Reviewer #4: Yes

3. Has the statistical analysis been performed appropriately and rigorously?

Reviewer #2: Yes

Reviewer #3: I don't know

Reviewer #4: Yes

4. Have the authors made all data underlying the findings in their manuscript fully available (please refer to the Data Availability Statement at the start of the manuscript PDF file)?

Reviewer #2: (No Response)

Reviewer #3: Yes

Reviewer #4: Yes

5. Is the manuscript presented in an intelligible fashion and written in standard English?

PLOS Digital Health does not copyedit accepted manuscripts, so the language in submitted articles must be clear, correct, and unambiguous. Any typographical or grammatical errors should be corrected at revision, so please note any specific errors here.

Reviewer #2: Yes

Reviewer #3: Yes

Reviewer #4: Yes

Reviewer #2: The revised manuscript demonstrates significant improvements based on reviewer feedback, particularly in clarifying methodological choices, addressing limitations, and expanding discussions on high-workload tasks and role reassessment.

With these revisions, the manuscript has strong potential to contribute valuable knowledge to the field of digital health nursing and inform strategies for optimizing nurse workloads in Virtual Care Centres.

Reviewer #3: Thank you for addressing all my comments

Reviewer #4: Thanks for revising the manuscript.

**Do you want your identity to be public for this peer review?** For information about this choice, including consent withdrawal, please see our Privacy Policy

Reviewer #2: No

Reviewer #3: No

Reviewer #4: No
